# Caspase-3, Caspase-8 and XIAP Gene Expression in the Placenta: Exploring the Causes of Spontaneous Preterm Labour

**DOI:** 10.3390/ijms24021692

**Published:** 2023-01-15

**Authors:** Vera Belousova, Oxana Svitich, Elena Timokhina, Irina Ignatko, Irina Bogomazova, Svetlana Pesegova, Tatiana Silaeva, Tatiana Kuzmina, Oxana Skorobogatova

**Affiliations:** 1I.M. Sechenov First Moscow State Medical University (Sechenov University), 119435 Moscow, Russia; 2Federal State Budgetary Scientific Institution «I. Mechnikov Research Institute of Vaccines and Sera», 105064 Moscow, Russia

**Keywords:** preterm birth, apoptosis, caspase

## Abstract

A better understanding of the pathogenesis of preterm birth (PTB) will allow us to lower the PTB rate, reducing perinatal morbidity and mortality. This article presents the hypothesis that premature placenta apoptosis could be a potential cause of PTB. We evaluated gene expression involved in apoptosis: caspase-3, caspase-8, and XIAP (X-linked inhibitor of apoptosis) in the placenta during pregnancy (n = 41), at the onset of preterm labour (n = 42), after preterm (n = 44) and term (n = 32) labour. We used RNA extraction, reverse transcription, and PCR. During pregnancy the gene expression of caspase-3 and caspase-8 is low, but XIAP is higher than the caspases. At the onset of preterm labour, we observed a significantly increased expression of both caspase-8 (10.7-fold, *p* < 0.01) and caspase-3 (2.5-fold, *p* < 0.01) and XIAP (3-fold; *p* < 0.05) compared with expression during pregnancy. Our study showed that during pregnancy, the expression of caspase genes in the placenta is low and probably controlled by high XIAP expression. At the onset of preterm labour, the expression of caspase genes increases sharply. This may initiate the onset of preterm labour.

## 1. Introduction

Around 15 million preterm babies are born globally every year; preterm birth (PTB) remains one of the major obstetrics challenges [1]. In Russia, the rate of preterm labour is around 6% [2]. These challenges are as much social as they are medical, with a high risk of infant death and numerous health complications [3]. A better understanding of the pathogenesis of preterm birth will increase prevention, allow us to carry out specific target therapy, and ultimately reduce the instances of PTB. Currently, the four most common causes leading to the onset of spontaneous PTB are the following [4]:Stress-induced premature activation of the maternal or foetal hypothalamic–pituitary–adrenal axis;Exaggerated inflammatory response/infection and/or an altered genital tract microbiome;Abruption (decidual hemorrhage);Pathologic uterine distention.

The need to understand the underlying causes and conditions leading to the spontaneous preterm uterine contractions remains a key issue in obstetrics. Currently, there are numerous studies aimed at revealing the mechanisms of the onset of PTB [5,6,7]. Considering that the placenta plays no role after birth, we made the decision to carry out a comparative study of placental apoptosis at term vs. preterm birth to determine if the intensive and premature activation of placental apoptosis could potentially trigger preterm labour.

Apoptosis is one of the processes describing the natural occurrence of cell death in the body [8]. The process occurs during various pathological conditions and is programmed and necessary for the organism’s cure or development [9]. Placenta apoptosis is as much a component of natural development as it is in other tissues. The role of apoptosis is well demonstrated in processes such as trophoblast invasion or remodelling of spiral arteries. However, the untimely development of apoptosis can lead to the development and/or progression of various pathological processes, including pregnancy complications. The role of apoptosis has been proven in contributing to spontaneous miscarriages [10], preeclampsia [11], placental insufficiency [12], and premature rupture of membranes [13,14]. It is also understood that the onset of apoptosis leads to infection-induced preterm labour [15]. However, there is no data on the role of placental apoptosis in leading to spontaneous PTB when an infectious etiology is excluded or not proven.

According to the literature, there are extrinsic and intrinsic pathways of apoptosis [16]. Since the role of inflammation in the uteroplacental complex has been proven to trigger PTB [17], even with the exclusion of infection [18], we assume that the apoptosis leading to PTB can be induced by TNF-α or Fas / Fas ligand. Therefore, it was decided to study the extrinsic pathway and caspase-8 as the initiating caspase. Both pathways of apoptosis converge on effector caspase-3, which we have also researched. The activation of caspases is regulated by the inhibitor of apoptosis proteins (IAP) [19]. We chose X-linked inhibitor of apoptosis (XIAP) for this research because it is an inhibitor of caspase-3.

The aim of our prospective study was to investigate the gene expression of caspase-8, caspase-3, and XIAP in the placenta during pregnancy, preterm and term labour.

## 2. Results

In Group 1 (term placenta before the onset of labour), gene expression of caspase-8 and caspase-3 was low (161 and 261), and XIAP expression was higher than the expression of caspase genes (328) (Figure 1a).

In Group 2 (placenta after the onset of preterm labour), we observed a 10.7-fold increasing expression of caspase-8 gene (161 and 1728; *p* < 0.01), 2.5-fold increasing expression of caspase-3 gene (261 and 650; *p* < 0.01), and 3-fold increasing expression of XIAP (328 and 1000; *p* < 0.01) (Figure 1b).

In Group 3 (placenta after preterm vaginal birth), we observed an 8-fold decrease in caspase-8 gene expression (1728 and 219; *p* < 0.01), the same expression level of caspase-3 (650 and 568; *p* > 0.05), and a 2.75-fold decrease in XIAP expression (1000 and 363; *p* < 0.05) when compared to the onset of PTB (Figure 1c).

In Group 4 (placenta after term vaginal birth), caspase-8 gene expression was at the same level when compared to preterm labour (248 and 219; *p* > 0.05). However, the caspase-3 gene expression was 1.85-fold higher (1050 and 568; *p* < 0.05), and XIAP gene expression was 2.63-fold lower than after preterm labour (138 and 363; *p* < 0.05) (Figure 1d).

By the end of the preterm labour, the expression of initiator caspase-8 decreases 8-fold compared to the onset of preterm labour (from 1728 to 219; *p* < 0.01) and becomes comparable with the level during pregnancy (219 and 161; *p* > 0.05).

The expression profile of effector caspase-3 is different from the profile of the initiator caspase-8. At the onset and at the end of PTB, the expression of caspase-3 remains at the same level (650 and 568; *p* > 0.05), but 2.5-fold higher than during pregnancy (650 and 261; *p* < 0.05). After a term vaginal birth, the expression of caspase-3 is 1.8 times higher than with a PTB (1050 and 568; *p* > 0.05). These expression changes are comparable to the XIAP expression profile. At the onset of a PTB, the gene expression of the apoptosis inhibitor increases 3-fold compared with the level during pregnancy (1000 and 328; *p* < 0.05). Moreover, in the end of PTB, XIAP expression remains 2.6-fold higher than in the end of term labour (363 and 138; *p* < 0.05).

During pregnancy, the gene expression of caspase-3 and caspase-8 is low, but XIAP is higher than the caspases (Figure 1a). At the onset of preterm labour, we observed a significantly increased expression of both caspase-8 (10.7-fold; *p* < 0.01) and caspase-3 (2.5-fold; *p* < 0.01) and XIAP (3-fold; *p* < 0.05) compared with expression during pregnancy (Figure 1b). After vaginal PTB, the expression of caspase-8 returns to a low level comparable to its expression during pregnancy, and the expression of caspase-3 and XIAP remains high (Figure 1c). After vaginal term birth, there is a very high expression of caspase-3 and low expression of caspase-8 and XIAP (Figure 1d).

## 3. Discussion

There are numerous ongoing PTB studies trying to establish the cause of the premature spontaneous onset of labour. Understanding of the pathogenesis of PTB will allow us to develop targeted treatment preventing or controlling this process. Currently, the risk factors for PTB have been identified [1,20]; some links of pathogenesis have been established [4]; hormonal changes, genetic, and immunological factors have been studied [4,5,6,21]; the role of pro-inflammatory cytokines and the polymorphism of their genes [22]; and the role of toll-like receptors have been proven [23]. However, there remains little understanding of the pathogenesis of spontaneous onset of the preterm uterine contractions.

Apoptosis occurs in all tissues, including placenta, and its role in the development of pregnancy complications such as preeclampsia, placental insufficiency, and spontaneous miscarriages has been proven. A number of studies have shown that in all these pathological conditions, the apoptosis is higher than in normal pregnancy.

The aim of our study was to evaluate caspase-3, caspase-8, and XIAP gene expression in placental tissue during pregnancy (Group 1—term placenta before the onset of labour), at the onset (Group 2—placenta after the onset of preterm labour), after spontaneous PTB of unknown etiology (Group 3—placenta after preterm vaginal birth), and after term vaginal delivery (Group 4—placenta after term vaginal birth). We studied the expression of the initiator caspase-8 and effector caspase-3 genes, and the expression of the XIAP gene.

During pregnancy (the time of active functioning of the placenta), the expression profile of the caspases’ and XIAP genes is as follows: the gene expression of both caspases is low (161 and 261), while the expression of the XIAP gene predominates (328). This ratio of expressions can probably be explained by the active functioning of the placenta during pregnancy. Apoptosis is at a low level and is actively controlled and suppressed by the inhibitor.

At the onset of PTB, we observed a significant increased expression of both caspase-8 (10.7-fold; *p* < 0.01) and caspase-3 (2.5-fold; *p* < 0.01) compared with expression during pregnancy. The data obtained suggest a high expression of caspases in the placenta at the onset of PTB. Thus, we observed the activation of the apoptosis at the onset of PTB. We are suggesting that the activation of apoptosis can lead to the premature onset of uterine contractions. At the same time, at the onset of PTB we also noted a significant increase in XIAP gene expression (3-fold; *p* < 0.01). We assume that the inhibitor XIAP is trying to suppress premature apoptosis in the placenta. This could potentially be a compensatory increase of the apoptosis inhibitor’s expression to suppress placental apoptosis and to stop the onset of PTB.

After a vaginal PTB, the expression of initiator caspase-8 returns to a low level comparable to its expression during pregnancy. The expression level of effector caspase-3 remains at the same high level as at the onset of PTB. However, after PTB the expression of caspase-3 is still lower than after term delivery (1.85-fold; *p* < 0.05). This fact is comparable to the expression of XIAP, in which the expression is high after PTB and very low after term labour (2.63-fold; *p* < 0.05).

After a vaginal term birth, the gene expression profile looks somewhat different; in the end of term labour, the gene expression of initiator caspase-8 is comparable with the level during pregnancy (248 and 161; *p* > 0.05), 4-fold times increasing gene expression of effector caspase-3 (1050 and 261; *p* < 0.05), and 2.4-fold decrease XIAP gene expression (138 and 328; *p* < 0.05) compared with pregnancy. Thus, at term birth, active apoptosis is observed in the placenta, with a low activity of its inhibition system.

Thus, at the onset of PTB, we observed the activation of apoptosis in the placenta, which may be the initiating moment in the development of premature uterine contractions. At the same time, we observe the XIAP trying to suppress the premature placental apoptosis, manifested by its high gene expression. High XIAP gene expression during PTB can be interpreted as an attempt to ‘correct’ the pathological process in the placental tissue, unsuccessfully. This hypothesis is also supported by the fact that during term labour, XIAP gene expression not only does not increase, but decreases. The term labour is a physiological process and the physiological termination of placental functioning. Therefore, an active programmed cell death occurs in the placenta and the inhibitor doesn’t try to suppress it.

## 4. Materials and Methods

Placental tissue (159) was obtained as a result of preterm (86) and term (73) delivery with patients’ consent.

Placental tissue from full-term pregnancies (total 73) formed the following:Group 1 (total 41): term placenta before the onset of labour (patients were delivered by term elective C-section);Group 4 (total 32): placenta after term vaginal birth with no complications.

Placental tissue from spontaneous preterm births (total 86), defined by spontaneous preterm onset of the uterine contractions at 26–34 weeks formed the following:Group 2 (total 42): placenta after the onset of preterm spontaneous labour (patients were delivered by preterm emergency C-section);Group 3 (total 44): placenta after preterm spontaneous vaginal birth with no complications.

Thus, all analysed PTB placentas belong to patients with idiopathic spontaneous PTB of unclear etiology.

All patients provided their informed consent for participation in the study. The study was conducted in accordance with the Declaration of Helsinki, and the study protocol was approved by the Sechenov University Ethics Committee (No. 10–17 of 16 November 2017).

### 4.1. RNA Extraction

A 5 g piece was cut off from the central part of the placenta immediately after delivery. It was immediately placed in 300 μL of RPMI-1640 medium with glutamine (PanEco) and gentamicin (100 μg/mL) and taken to the lab for processing.

Placental tissue was homogenized for RNA extraction using the AmpliPrim RIBO—sorb kit of InterLabService (Moscow, Russia) as outlined by the manufacturer. A lysing solution was added to the placental homogenizate, and RNA was isolated by affinity sorption on silica gel particles using the AmpliPrim RIBO—sorb kit of InterLabService (Russia) strictly in accordance with the protocol.

### 4.2. Reverse Transcription

The isolated RNA was used as the template for cDNA synthesis in reverse transcription reactions using the OT-1 Reverse Transcription kit (Syntol, Russia).

Briefly, the reaction Mix-1 contained 4 μL of RNA, 10 nmoles of random hexamers, and 10 nmoles of primer (Synthol, Russia), and was incubated at 75 °C for 3 min and afterwards incubated for 30 min on ice. Mix-2 was prepared and contained dNTP (at a concentration of 1 mM each), 40 units of RNase inhibitor, and 100 units of M-MLV revertase and buffer for reverse transcription from the kit. Then Mix-2 was added to Mix-1 and incubated at 37 °C for 1 h, then incubated at 94 °C for 10 min. The samples with cDNA were stored at −70 °C.

### 4.3. Polymerase Chain Reaction (PCR)

Primer sequences for caspase-3, caspase-8, and XIAP were selected using Vector NTI 8.0 to analyze the gene sequences obtained from the electronic database GenBank (see Table 1).

The PCR mix included: Taq pol. 5 mL/mL, forward primer 10 nm, reverse primer 10 nm (see Table 1), 1.2 mm dNTP, and buffer for PCR. The total volume of the PCR mix was 30 µL, 3 µL of which was cDNA. For PCR it is used “Kit for PCR-RV in the presence of intercalating dye SYBR Green I (Buffer B)” (Synthol, Russia). All of the production reaction was carried out according to the recommendations of the manufacturer. The PCR program comprised the following: 94 °C, 2 min, (94 °C—15 s, 60 °C—15 s, 72 °C—30 s) 40 cycles. For the PCR reaction, a thermocycler DT-96 (DNA-technology, Moscow, Russia) was used. To standardize the results, the expression of the housekeeping gene GAPDH was used.

Data was analyzed using SPSS Version 20, Excel.

All data were analyzed and defined using the Mann–Whitney test with *p* < 0.05 considered significant.

## 5. Conclusions

During pregnancy, we have observed a low activity of the apoptosis process in the placenta, in part due to the active expression of the apoptosis inhibitor gene. In term labour, we see a high expression of caspase genes, while the expression of apoptosis inhibitor genes is at a low level.

In preterm birth, there is a high expression of the initiator and effector caspase genes and their inhibitor; this could, we suggest, be the initiating moment in the development of PTB. The XIAP inhibitor tries to suppress the prematurely launched apoptosis process in premature labour, which is manifested by its high expression. Changes in XIAP expression in PTB may indicate an attempt to correct the pathological process in placental cells in PTB.

Our research has enabled us to establish the role of gene expression of caspases and XIAP in the placenta in spontaneous preterm labour. The next phase of this research will study apoptotic proteins in the placenta, with the objective to develop etiopathogenetic treatment aimed at the prevention and suppression of preterm labour.

We suggest the preterm placental apoptosis may be the trigger of sterile inflammation in the utero-placental complex. As a result of the premature death of placental cells, damage-associated molecular patterns (DAMPs) are released, and by the activation of toll-like receptors could lead to the sterile inflammation in the utero-placental complex and the active production of prostaglandins. Studying and understanding the pathogenesis of spontaneous preterm labour can help to prevent and choose tocolytic therapy.

## Figures and Tables

**Figure 1 ijms-24-01692-f001:**
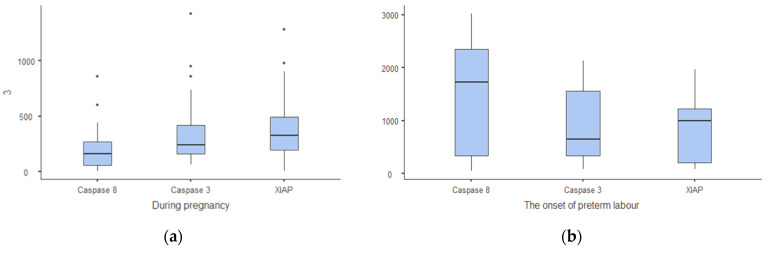
(**a**–**d**) Gene expression of caspase-8, caspase-3, and XIAP in placenta during pregnancy, at the onset of PTB and after PTB and term labour.

**Table 1 ijms-24-01692-t001:** Primer sequences for caspase-3, caspase-8, and XIAP.

Name of Gene	GenBank	Name of Oligonucleotide	Sequence
Caspase-3	NM_032991	Cas3_f	atg-gaa-gcg-aat-caa-tgg-act-ct
		Cas3_r	ctg-cat-cga-cat-ctg-tac-cag-a
Caspase-8	NM_001228	Cas8_f	cct-ccc-tca-agt-tcc-tga-gcc-t
		Cas8_r	ttc-cct-ttc-cat-ctc-ctc-ctt-tct
XIAP	NG_007264	XIAP_f1	att-gcc-ttt-cct-gct-aca-tt
		XIAP_r	cca-ttc-aac-acc-tgt-gta-at

## Data Availability

Not applicable.

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
