# Peer review of "Caspase-3, Caspase-8 and XIAP Gene Expression in the Placenta: Exploring the Causes of Spontaneous Preterm Labour"

_ijms, 2023, doi:10.3390/ijms24021692_

Round 1

Reviewer 1 Report

This study reports significantly higher expression of apoptotic genes at the onset of preterm labour compared to the expression during pregnancy. I have the following comments:

1.       The authors should indicate the rate of preterm labour in Russia (in the introduction section).

2.       It is not clear whether this is a prospective or retrospective study.

3.       What were the inclusion criteria? The authors should further define idiopathic preterm labour and a flowchart of patients’ inclusion should be added (when did the study take place? How many patients with preterm labour delivered in authors’ institution during the study? Which patients were included and excluded?)

4.       Were the patients who had conisation prior to pregnancy included?

5.       A table with patients’ characteristics should be added (age, gestational age…)

6.       A possible clinical application of these findings should be added to the conclusions section.

Author Response

Thank you for the work you have done and comments which can help to improve our manuscript.

  1. The authors should indicate the rate of preterm labour in Russia (in the introduction section) - In Russia the rate of preterm labour is around 6% (we clarified this in the manuscript)
  2. It is not clear whether this is a prospective or retrospective study – Our study was prospective.( we clarified this in the manuscript)
  3. What were the inclusion criteria? The authors should further define idiopathic preterm labour and a flowchart of patients’ inclusion should be added (when did the study take place? How many patients with preterm labour delivered in authors’ institution during the study? Which patients were included and excluded?)

Women with suspected intraamniotic and/or other infection (as chorioamnionitis), after cervical conisation, premature rupture of the membranes, preeclampsia and its complications (eclampsia, placental abruption, HELLP syndrome, etc.), placental insufficiency, placenta previa, foetal anomalies, medical complications (such as gestational diabetes mellitus, severe chronical somatic diseases), multiple gestation, pregnancy after assisted reproductive technology and those who had any surgical procedures during pregnancy were excluded.( we clarified this in the manuscript)

Idiopathic preterm labour is preterm spontaneous labour of unclear etiology (we clarified this in the manuscript)

  1. Were the patients who had conisation prior to pregnancy included? –no, such patients were excluded.
  2. A table with patients’ characteristics should be added (age, gestational age…) - our study included healthy women of similar age, and therefore we did not see the need for a table with patient characteristics.
  3. A possible clinical application of these findings should be added to the conclusions section. - We suggest the preterm placental apoptosis may be the trigger of sterile inflammation in the utero-placental complex. As a result of premature death of placental cells, damage-associated molecular patterns (DAMPs) are released and by the activation of toll-like receptors could lead to the sterile inflammation in the utero-placental complex and the active production of prostaglandins. Studying and understanding the pathogenesis of spontaneous preterm labour can help to prevent and choice tocolytic therapy. (we include this part in conclusion)

Reviewer 2 Report

Articles "Caspase-3, caspase-8 and XIAP gene expression in the placenta: exploring the causes of spontaneous preterm labor" has all the characteristics of original research. The introduction is suitable, the material and methods are adequate, the presentation of the results is suitable for the given goals. The discussion and conclusion are adequate with recent references. In conclusion, the authors state "our study showed that during pregnancy the expression of caspase genes in the placenta is low and probably controlled by high XIAP expression. At the onset of preterm labor the expression of caspase genes increases sharply. This may initiate the onset of preterm labor." According to the above, the paper is suitable for publication with this topic for this journal.

Author Response

Thank you so much for such a high appreciation of our manuscript!

Round 2

Reviewer 1 Report

I have no further comments